# Late Pleistocene human genome suggests a local origin for the first farmers of central Anatolia

Michal Feldman[1], Eva Fernández-Domínguez[2], Luke Reynolds[3], Douglas Baird[4], Jessica Pearson[4], Israel Hershkovitz[5,6], Hila May[5,6], Nigel Goring-Morris[7], Marion Benz[8], Julia Gresky[9], Raffaela A. Bianco[1], Andrew Fairbairn[10], Gökhan Mustafaoğlu[11], Philipp W. Stockhammer[1,12], Cosimo Posth [1], Wolfgang Haak[1], Choongwon Jeong [1] & Johannes Krause [1]

Anatolia was home to some of the earliest farming communities. It has been long debated whether a migration of farming groups introduced agriculture to central Anatolia. Here, we report the first genome-wide data from a 15,000-year-old Anatolian hunter-gatherer and from seven Anatolian and Levantine early farmers. We find high genetic continuity (~80–90%) between the hunter-gatherers and early farmers of Anatolia and detect two distinct incoming ancestries: an early Iranian/Caucasus related one and a later one linked to the ancient Levant. Finally, we observe a genetic link between southern Europe and the Near East predating 15,000 years ago. Our results suggest a limited role of human migration in the emergence of agriculture in central Anatolia.

[1] Max Planck Institute for the Science of Human History (MPI-SHH), Kahlaische Strasse 10, 07745 Jena, Germany. [2] Department of Archaeology, Durham University, DurhamSouth Road, DH1 3LE, UK. [3] School of Natural Sciences and Psychology, Liverpool John Moores University, Byrom Street, Liverpool L3 3AF, UK. [4] Department of Archaeology, Classics and Egyptology, University of Liverpool, 8–14 Abercromby Square, Liverpool L69 7WZ, UK. [5] Department of Anatomy and Anthropology, The Dan David Center for Human Evolution and Biohistory Research and The Shmunis Family Anthropology Institute, Sackler Faculty of Medicine, Tel Aviv University, Post Office Box 39040, Tel Aviv 6997801, Israel. [6] The Steinhardt Museum of Natural History, Tel Aviv University, Post Office Box 39040, Tel Aviv 6997801, Israel. [7] Department of Prehistory, Institute of Archaeology, The Hebrew University of Jerusalem, Jerusalem 919051, Israel. [8] Department of Near Eastern Archaeology, Free University Berlin, Fabeckstrasse 23-25, 14195 Berlin, Germany. [9] Department of Natural Sciences, German Archaeological Institute, Im Dol 2-6, 14195 Berlin, Germany. [10] School of Social Science, The University of Queensland, Michie Building, St Lucia, Brisbane, QLD, Australia. [11] Department of Archaeology, Zonguldak Bülent Ecevit University, Incivez, 67100 Zonguldak, Turkey. [12] Institut für Vor- und Frühgeschichtliche Archäologie und Provinzialrömische, Archäologie Ludwig-Maximilians-Universität München München, Schellingstrasse 12, 80799 München, Germany. Correspondence and requests for materials should be addressed to E.F.-D. (email: eva.fernandez@durham.ac.uk) or to C.J. (email: jeong@shh.mpg.de) or to J.K. (email: krause@shh.mpg.de)

The practice of agriculture began in the Fertile Crescent of Southwest Asia as early as 10,000 to 9000 BCE. Subsequently, it spread across western Eurasia while increasingly replacing local hunting and gathering subsistence practices, reaching central Anatolia by c. 8300 BCE[1–3].

Recent genetic studies have shown that in mainland Europe, farming was introduced by an expansion of early farmers from Anatolia that replaced much of the local populations[4,5]. Such mode of spread is often referred to as the demic diffusion model. In contrast, in regions of the Fertile Crescent such as the southern Levant and the Zagros Mountains (located between present-day eastern Iraq and western Iran), the population structure persists throughout the Neolithic transition[6], indicating that the hunter-gatherers of these regions locally transitioned to a food-producing subsistence strategy.

Central Anatolia has some of the earliest evidence of agricultural societies outside the Fertile Crescent[3] and thus is a key region in understanding the early spread of farming. While archeological evidence points to cultural continuity in central Anatolia[3], due to the lack of genetic data from pre-farming individuals, it remains an open question whether and to what scale the development of the Anatolian Neolithic involved immigrants from earlier farming centers admixing with the local hunter-gatherers.

Likewise, pre-farming genetic links between Near-Eastern and European hunter-gatherers are not well understood, partly due to the lack of hunter-gatherer genomes from Anatolia. Genetic studies have suggested that ancient Near-Eastern populations derived a substantial proportion of their ancestry from a common outgroup of European hunter-gatherers and East Asians[4,6,7]. This deeply branching ancestry often referred to as Basal Eurasian likely diverged from other Eurasians before the latter received Neanderthal gene flow[6]. Interestingly, a previous study reported that European hunter-gatherers younger than 14,000 years ago tend to show an increased affinity with present-day Near-Easterners compared to older European hunter-gatherers[8], although how this affinity formed is not well understood.

Here, we report new genome-wide data from eight prehistoric humans (Fig. 1a, Table 1, and Supplementary Table 1), including the first Epipaleolithic Anatolian hunter-gatherer sequenced to date (labeled AHG; directly dated to 13,642–13,073 cal BCE, excavated from the site of Pınarbaşı, Turkey), five early Neolithic Aceramic Anatolian farmers (labeled AAF; c. 8300–7800 BCE, one directly dated to 8269–8210 cal BCE[3], from the site of Boncuklu, Turkey), adding to previously published genomes from this site[9], and two Early Neolithic (PPNB) farmers from the southern Levant (one labeled KFH2, directly dated to c. 7700–7600 cal BCE, from the site of Kfar HaHoresh, Israel; and the second labeled BAJ001, c. 7027–6685 cal BCE, from the site of Ba'ja, Jordan). These data comprise a genetic record stretching from the Epipaleolithic into the Early Holocene, spanning the advent of agriculture in the region.

We find that the AHG is genetically distinct from other reported late Pleistocene populations. We reveal that Neolithic Anatolian populations derive a large fraction of their ancestry from the Epipaleolithic Anatolian population, suggesting that farming was adopted locally by the hunter-gatherers of central Anatolia. We also detect distinct genetic interactions between the populations of central Anatolia and earlier farming centers to the east, during the late Pleistocene/early Holocene and describe a genetic link with European hunter-gatherers that predates 15,000 years ago.

## Results

### Genetic continuity and detected admixtures in Anatolia. We extracted DNA from the ancient human remains and prepared it for next-generation sequencing[10,11], which resulted in human

DNA yields lower than 2% (Supplementary Data 1), comparable with low DNA preservation previously reported in the region[6,9]. To generate genome-wide data despite the low DNA yields, we performed in-solution DNA enrichment targeting 1.24 million genome-wide single-nucleotide polymorphisms (SNPs) ("1240k capture")[12], which resulted in 129,406 to 917,473 covered SNPs per individual. We estimated low mitochondrial contamination levels for all eight individuals (1–6%; see Methods and Supplementary Table 2) and could further test the males for nuclear contamination, resulting in low estimates (0.05–2.23%; Supplementary Table 2). For population genetic analyses, we merged genotype data of the new individuals with previously published datasets from 587 ancient individuals and 254 present-day populations (Supplementary data 2).

To estimate how the ancient individuals relate to the known west Eurasian genetic variation, we projected them onto the top two dimensions (PC1, PC2) of present-day principal component analysis (PCA)[6] (Fig. 1b). Strikingly, the AHG individual is positioned near both AAF and later Anatolian Ceramic farmers[12] (7000–6000 cal BCE). These three prehistoric Anatolian populations (AHG, AAF, and ACF), representing a temporal transect spanning the transition into farming, are positioned along PC1 between Mesolithic western European hunter-gatherers (WHG)[4,7,12] who are at one extreme of PC1 and Levantine Epipaleolithic Natufians[6] who are at the other. Along PC2, ancient Anatolians, WHG, and Natufians have similar coordinates. The newly reported Levantine Neolithic farmers (BAJ001 and KFH2) are positioned near the previously published Levantine Neolithic farmers[6] (Supplementary Note 2). In ADMIXTURE analysis AHG, AAF, and ACF are inferred as a mixture of two components that are each maximized in Natufians and WHG, consistent with their intermediate positions along PC1 in PCA (Supplementary Figure 1).

Inspired by our qualitative observations in PCA and ADMIXTURE analyses, we applied formal statistical frameworks to describe the genetic profiles of the three Anatolian populations and to test and model genetic differences between them. We first characterized the ancestry of AHG. As expected from AHG's intermediate position on PCA between Epipaleolithic/Neolithic Levantines and WHG, Patterson's D-statistics[13] of the form $D$ (AHG, WHG; Natufian/Levant_N, Mbuti) $\geq 4.8$ SE (standard error) and $D$ (AHG, Natufian/Levant_N; WHG, Mbuti) $\geq 9.0$ SE (Supplementary Table 3) indicate that AHG is distinct from both the WHG and Epipaleolithic/Neolithic Levantine populations and yet shares extra affinity with each when compared to the other. Then, we applied a qpAdm-based admixture modeling to integrate these D- statistics. qpAdm is a generalization of $D/f_4$-statistics that test whether the target population and the admixture model (i.e., a linear combination of reference populations) are symmetrically related to multiple outgroups[13]. By doing so, it tests whether the proposed admixture model is adequate to explain the target gene pool and provides admixture coefficient estimates. We find an adequate two-way admixture model ($\chi^2 p = 0.158$), in which AHG derives around half of his ancestry from a Neolithic Levantine-related gene pool ($48.0 \pm 4.5\%$; estimate $\pm 1$ SE) and the rest from the WHG-related one (Supplementary Tables 4, 5). While these results do not suggest that the AHG gene pool originated as a mixture of Levant_N and WHG, both of which lived millennia later than AHG, it still robustly supports that AHG is genetically intermediate between WHG and Levant_N. This cannot be explained without gene flow between the ancestral gene pools of those three groups. This supports a late Pleistocene presence of both Near-Eastern and European hunter-gatherer-related ancestries in central Anatolia. Notably, this genetic link with the Levant pre-dates the advent of farming in this region by at least five millennia.

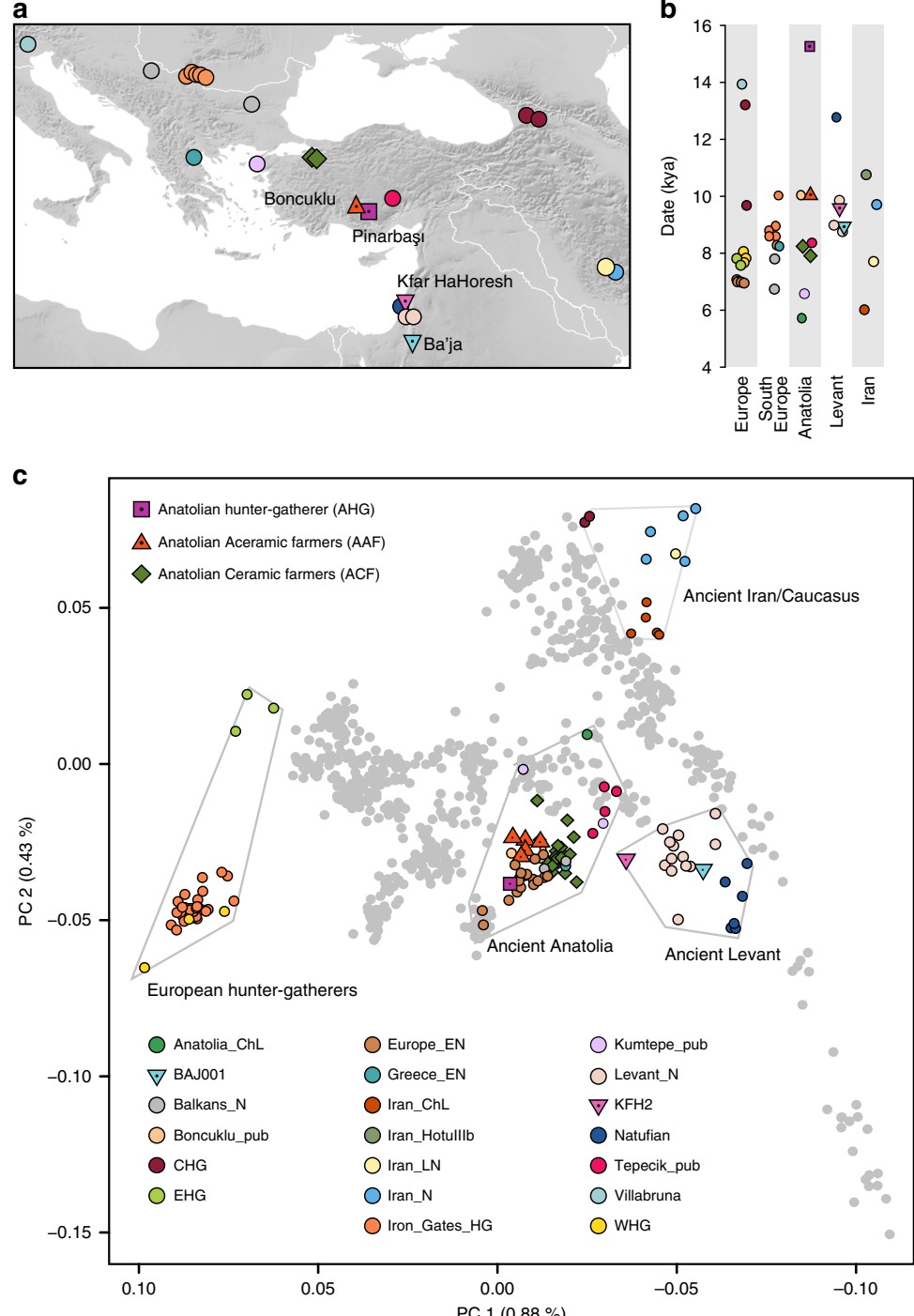

**Fig. 1** Location, age, and principal component analysis (PCA) of analyzed individuals. **a** Locations of newly reported and selected published genomes. Archeological sites from which new data are reported are annotated. Symbols for the analyzed groups are annotated in **c**. **b** Average ages of ancient groups. **c** Ancient genomes (marked with color-filled symbols) projected onto the principal components computed from present-day west Eurasians (gray circles) (Supplementary Figure 8). The geographic location of each ancient group is marked in **a**. Ancient individuals newly reported in this study are additionally marked with a black dot inside the symbol. Source data are provided as a Source Data file

In turn, AAF are slightly shifted on PC2 compared to AHG, to the direction where ancient and modern Caucasus and Iranian groups are located. Likewise, when compared to AHG by $D(AAF, AHG; test, Mbuti)$, the AAF early farmers show a marginal excess affinity with early Holocene populations from Iran or Caucasus and with present-day south Asians, who have also been genetically linked with Iranian/Caucasus ancestry[14,15] (e.g., $D =$ 2.3 and 2.7SE for CHG and Vishwabrahmin, respectively; Fig. 2a,

Supplementary Figures 2, 3, and Supplementary Data 3). Accordingly, a mixture of AHG and Neolithic Iranians provides a good fit to AAF in our *qpAdm* modeling ($\chi^2 p = 0.296$), in which AAF derive most of their ancestry (89.7 ± 3.9%) from a population related to AHG (Supplementary Tables 4 and 6). A simpler model without contribution from Neolithic Iranians (i.e., modeling AAF as a sister clade of AHG) shows a significant reduction in model fit ($\chi^2 p = 0.014$). This suggests a long-term

**Table 1 An overview of ancient genomes reported in this study**

| ID | Library name | Analysis group | Estimated date | Site | Sampled tissue | Total sequenced reads (×10⁶) | Human DNA (%) | Mean coverage (fold) | Genetic sex | mt | Ychr |
|---|---|---|---|---|---|---|---|---|---|---|---|
| ZBC | IPB001.B/C0101 | AHG | 13,642–13,073 cal BCE | Pınarbaşı | Intermediate phalanx | 126.7 | 33 | 2.9 | Male | K2b | C1a2 |
| ZHAG | BON004.A0101 | AAF | 8300–7800 BCE | Boncuklu | Petrous | 92.0 | 38 | 1.48 | Female | N1a1a1 | |
| ZMOJ | BON014.A0101 | AAF | 8300–7800 BCE | Boncuklu | third molar | 77.9 | 27 | 0.8 | Male | K1a | C |
| ZKO | BON001.A0101 | AAF | 8300–7800 BCE | Boncuklu | Petrous | 84.8 | 31 | 0.9 | Male | U3 | G2a2b2b |
| ZHJ | BON024.A0101 | AAF | 8300–7800 BCE | Boncuklu | Third molar | 87.7 | 38 | 0.76 | Female | U3 | |
| ZHAJ | BON034.A0101 | AAF | 8269–8210 cal BCE | Boncuklu | Petrous | 75.4 | 30 | 0.69 | Female | U3 | |
| KFH2 | KFH002.A0101 | Levant_Neol | 7712–7589 cal BCE | Kfar HaHoresh | Petrous | 342.0 | 8 | 0.16 | Female | N1a1b | |
| BAJ001 | BAJ001.A0101 | Levant_Neol | 7027–6685 cal BCE | Ba'ja | Petrous | 17.3 | 45 | 0.75 | Female | N1b1a | |

For each individual the analysis group is given (AHG = Anatolian hunter-gatherer; AAF = Anatolian Aceramic farmers; Levant_Neol = Levantine early farmer). When ¹⁴C dating results are available, the date is given in cal BCE in 2-sigma range, otherwise a date based on the archeological context is provided (detailed dating information is provided in Supplementary Note 1 and Supplementary Table 1). The proportion of human DNA and the mean coverage on 1240k target sites in the "1240k" enriched libraries are given. Uniparental haplogroups (mt = mitochondrial; Ychr = Y chromosome) are listed. Detailed information on the uniparental analysis can be found in Supplementary Note 1 and Supplementary Data 6

genetic stability in central Anatolia over five millennia despite changes in climate and subsistence strategy. The additional Neolithic Iranian-related ancestry ($10.3 \pm 3.9\%$) presumably diffused into central Anatolia during the final stages of the Pleistocene or early Holocene, most likely via contact through eastern Anatolia. This provides evidence of interactions between eastern and central Anatolia in the Younger Dryas or the first millennium of the Holocene, currently poorly documented archeologically.

In contrast, we find that the later ACF individuals share more alleles with the early Holocene Levantines than AAF do, as shown by positive $D(ACF, AAF; Natufian/Levant\_N, Mbuti) \geq 3.8$ SE (Fig. 2b, Supplementary Figures 4, 5, and Supplementary Data 3). Ancient Iran/Caucasus populations and contemporary South Asians do not share more alleles with ACF ($|D| < 1.3$ SE). Likewise, qpAdm modeling suggests that the AAF gene pool still constitutes more than 3/4 of the ancestry of ACF 2000 years later ($78.7 \pm 3.5\%$; Supplementary Tables 4 and 7) with additional ancestry well modeled by the Neolithic Levantines ($\chi^2 p = 0.115$) but not by the Neolithic Iranians ($\chi^2 p = 0.076$; the model estimated infeasible negative mixture proportions) (Supplementary Tables 4 and 7). These results suggest gene flow from the Levant to Anatolia during the early Neolithic. In turn, Levantine early farmers (Levant_Neol) that are temporally intermediate between AAF and ACF could be modeled as a two-way mixture of Natufians and AHG or AAF ($18.2 \pm 6.4\%$ AHG or $21.3 \pm 6.3\%$ AAF ancestry; Supplementary Tables 4 and 8 and Supplementary Data 4), confirming previous reports of an Anatolian-like ancestry contributing to the Levantine Neolithic gene pool[6]. These two distinct detected gene flows support a reciprocal genetic exchange between the Levant and Anatolia during the early stages of the transition to farming.

**Genetic links between Pleistocene Europe and the Near East.** AHGs experienced climatic changes during the last glaciation[16] and inhabited a region that connects Europe to the Near East. However, pre-Neolithic interactions between Anatolia and Southeastern Europe are so far not well documented archeologically. Interestingly, a previous genomic study showed that present-day Near Easterners share more alleles with European hunter-gatherers younger than 14,000BP ("Later European HG") than with older ones ("Earlier European HG")[8]. With ancient genomic data available, we could directly compare the genetic affinity of European hunter-gatherers with Near-Eastern hunter-gatherers (AHG and Natufian) using the D-statistic of the form $D(European \ hunter-gatherers, Kostenki14; AHG/Natufian, Mbuti)$. We compared the European hunter-gatherers to the 37 thousand-year-old individual Kostenki14[8,17] representing the oldest available European genome with genetic affinity to later European hunter-gatherers (Fig. 3a and Supplementary Data 5). As is the case for present-day Near Easterners, this statistic is significantly positive for all European hunter-gatherers younger than 14,000BP. Most of the Later European HGs belong to a largely homogeneous gene pool referred to as the "Villabruna cluster,"[8] named after its oldest available member from an Epigravettian site in northern Italy. Our results suggest that the non-Basal Eurasian ancestry of ancient Anatolians and Levantines derived from a gene pool related to the Villabruna cluster prior to its expansion within Europe observed after 14,000BP.

Among the Later European HG, recently reported Mesolithic hunter-gatherers from the Balkan peninsula, which geographically connects Anatolia and central Europe ("Iron Gates HG")[18], show the highest genetic affinity to AHG and the second highest one to Natufians, as shown in the positive statistic $D(Iron\_Gates\_HG, European \ hunter-gatherers; AHG/Natufian, Mbuti)$ (Supplementary Figures 6 and 7). This affinity is surprising considering that Iron Gates HG have been previously modeled as a mixture of WHG (~85%) and eastern European hunter-gatherers (EHG; ~15%)[18], the latter of which shares a much lower affinity with ancient Near Easterners in respect to other European HG (Fig. 3a). Since the previously reported WHG and EHG model did not fit well ($\chi^2 p = 0.0003$) and since Iron Gates HG harbored Near-Eastern-like mitochondrial groups, an affinity with Anatolians beyond the WHG + EHG model has been hypothesized[18]. Accordingly, we find that Iron Gates HG can be modeled as a three-way mixture of Near-Eastern hunter-gatherers ($25.8 \pm 5.0\%$ AHG or $11.1 \pm 2.2\%$ Natufian), WHG ($62.9 \pm 7.4\%$ or $78.0 \pm 4.6\%$, respectively) and EHG ($11.3 \pm 3.3\%$ or $10.9 \pm 3\%$,

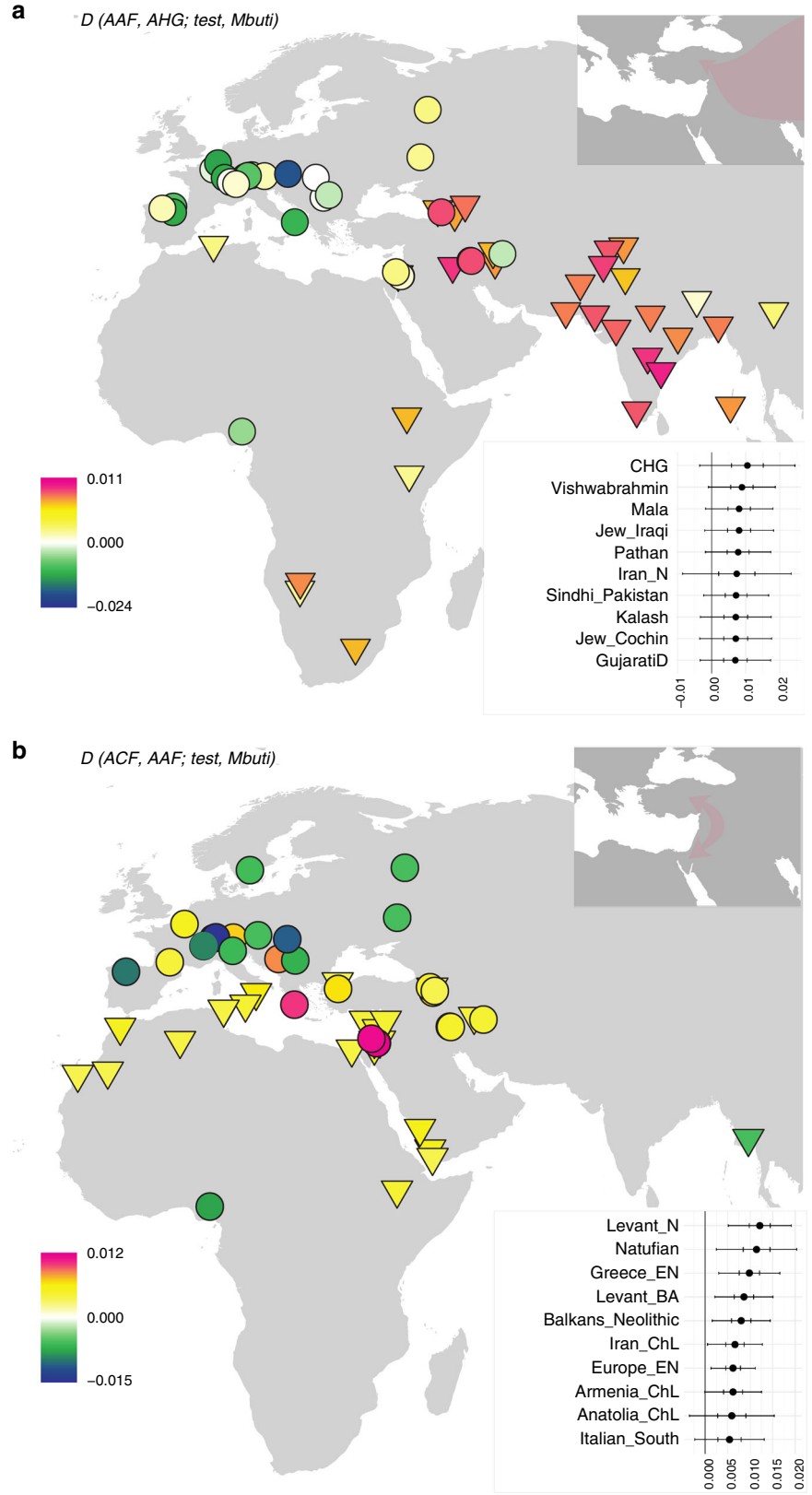

respectively); ($\chi^2 p = 0.308$ and $\chi^2 p = 0.589$ respectively; Supplementary Tables 4 and 9).

To further test the model of Near-Eastern gene flow into the ancestors of Iron Gates HG as an explanation of the extra affinity between them, we utilized the Basal Eurasian ancestry that was widespread in early Holocene and late Pleistocene Near-Eastern

populations and their descendants but undetectable in European hunter-gatherers[8], as a tracer for gene flow from the Near East. To estimate the Basal Eurasian ancestry proportion ("α"), we followed a previously established *qpAdm*-based approach that uses an African reference (the ancient Ethiopian *Mota* genome[19]) as a proxy[6] (Supplementary Table 10). We estimated α to be 24.8

**Fig. 2** Differences in genetic affinities between the ancient Anatolian populations. We plot the highest and lowest 40 values of D(population 1, population 2; test, Mbuti) on the map. Circles mark ancient populations and triangles present-day ones. "Test" share more alleles with population 1 when values are positive and with population 2 when negative. The detected gene flow direction is illustrated in the upper schematics; the illustrated rout represents the shortest one between the proximate source and the target and should not be interpreted as the historic rout of the gene flow. The statistics and SEs are found in Supplementary Figures 2–5 and Supplementary Data 3. **a** Early Holocene Iranian and Caucasus populations, as well as present-day South Asians, share more alleles with Aceramic Anatolian farmers (AAF) than with Anatolian hunter-gatherers (AHG), measured by positive D(AAF, AHG; test, mbuti). The top 10 values with ±1 and ±3SE are shown in the upper box. **b** Ancient Levantine populations share more alleles with Anatolian Ceramic farmers (ACF) than with AAF, measured by positive D(ACF, AAF; test, Mbuti). The top 10 values with ±1 and ±3 SE are shown in the lower box. Source data are provided as a Source Data file

± 5.5% in AHG and 38.5 ± 5.0% in Natufians (Fig. 3b, Supplementary Table 10), consistent with previous estimates for the latter[6]. In turn, the Iron Gates HG could be modeled without any Basal Eurasian ancestry or with a non-significant proportion of 1.6 ± 2.8% when forced to have it as a third source (Fig. 3b and Supplementary Table 10). In contrast to the above direct estimate, the three-way admixture model of WHG + EHG + AHG predicts $\alpha = 6.4 \pm 1.9\%$ for Iron Gates, calculated as (% AHG in Iron Gates HG) × ($\alpha$ in AHG), suggesting that unidirectional gene flow from the Near East to Europe alone may not be sufficient to explain the excess affinity between the Iron Gates HG and the Near-Eastern hunter-gatherers. Thus, it is plausible to assume that prior to 15,000 years ago there was either a bidirectional gene flow between populations ancestral to Southeastern Europeans of the early Holocene and those ancestral to Anatolians of the Late Glacial or a genetic influx from the populations ancestral to Southeastern Europeans into the Near East.

**Uniparental markers and phenotypic analysis.** The uniparental marker analysis placed AHG within the mitochondrial sub-haplogroup K2b and within the Y-chromosome haplogroup C1a2, both rare in present-day Eurasians (Table 1 and Supplementary Data 6). Mitochondrial Haplogroup K2 has so far not been found in Paleolithic hunter-gatherers[20]. However, Y-haplogroup C1a2 has been reported in some of the earliest European hunter-gatherers[8,17,21]. The early farmers belong to common early Neolithic mitochondrial (N1a, U3 and K1a) and Y chromosome types (C and G2a), with the exception of the Levantine BAJ001, which represents the earliest reported individual carrying the mitochondrial N1b group (Table 1 and Supplementary Data 6).

We examined alleles related to phenotypic traits in the ancient genomes (Supplementary Data 7). Notably, three of the AAF carry the derived allele for rs12193832 in the HERC2 (hect domain and RLD2) gene that is primarily responsible for lighter eye color in Europeans[22]. The derived allele is observed as early as 14,000–13,000 years ago in individuals from Italy and the Caucasus[8,23], but had not yet been reported in early farmers or hunter-gatherers from the Near East.

## Discussion

By analyzing genome-wide-data from pre-Neolithic and early Neolithic Anatolians and Levantines, we describe the demographic developments leading to the formation of the Anatolian early farmer population that later replaced most of the European hunter-gatherers and represents the largest ancestral component in present-day Europeans[4,5].

We report a long-term persistence of the local AHG gene pool over seven millennia and throughout the transition from foraging to farming. This demographic pattern is similar to those previously observed in earlier farming centers of the Fertile Crescent[6] and differs from the pattern of the demic diffusion-based spread of farming into Europe[4,5]. Our results provide a genetic support

for archeological evidence[3], suggesting that Anatolia was not merely a stepping stone in a movement of early farmers from the Fertile Crescent into Europe, but rather a place where local hunter-gatherers adopted ideas, plants, and technology that led to agricultural subsistence.

Interestingly, while we observe a continued presence of the AHG-related gene pool throughout the studied period, a pattern of genetic interactions with neighboring regions is evident from as early as the Late Pleistocene and early Holocene. In addition to the local genetic contribution from earlier Anatolian populations, Anatolian Aceramic farmers inherit about 10% of their genes from a gene pool related to the Neolithic Iran/Caucasus while later ACF derive about 20% of their genes from another distinct gene pool related to the Neolithic Levant.

Wide temporal gaps between available genomes currently limit our ability to distinguish the mode of transfer. Obtaining additional genomic data from these regions as well as from geographically intermediate populations of eastern Anatolia and the greater Mesopotamia region could help determine how these genetic changes happened in central Anatolia: for example, whether by a short-term massive migration or a low-level background gene flow in an isolation by distance manner.

To the west, we observe a genetic link between the Anatolian and European Pleistocene hunter-gatherers, which extends the temporal frame of the previously reported genetic affinity between late Pleistocene Europeans and present-day Near-Eastern populations[8]. Especially, Mesolithic Southeastern European hunter-gatherers (Iron Gates HG) show a strong genetic affinity with AHG. Our analysis on their Basal Eurasian ancestry proportions, although limited in resolution, suggests that a Near-Eastern gene flow from AHG into the ancestors of Iron Gates HG may not be sufficient to explain this affinity. Two additional scenarios, both involving gene flow from the ancestors of Iron Gates HG to the ancestors of AHG, can help explain the extra affinity between Iron Gates HG and AHG. One assumes a secondary gene flow from Southeastern Europe to Anatolia after the initial formation of the Near-Eastern gene pool as a mixture of the Basal Eurasian and the Villabruna-related gene pools. The other assumes that Iron Gates HG are indeed the most closely related group among European hunter-gatherers to the Villabruna-related ancestry in ancient Near Easterners. Further sampling in Anatolia and Southeastern Europe is needed to specify the spatiotemporal extent of the genetic interactions that we observe.

## Methods

**aDNA analysis.** We extracted and prepared DNA for next-generation sequencing in two different dedicated ancient DNA (aDNA) facilities (Liverpool and Jena).

In Liverpool, UK, sampling and extraction steps for the individuals from Pınarbaşı and Boncuklu were carried out in the aDNA labs at the Liverpool John Moores University. The outer layer of the bone was removed using powdered aluminum oxide in a sandblasting instrument. Then, the bone was ultraviolet (UV) irradiated for 10 min on each side and ground into fine powder using a cryogenic grinder Freezer/Mill. DNA was extracted from 100 mg of bone powder following an established protocol[10]. The extraction included incubation of the bone powder

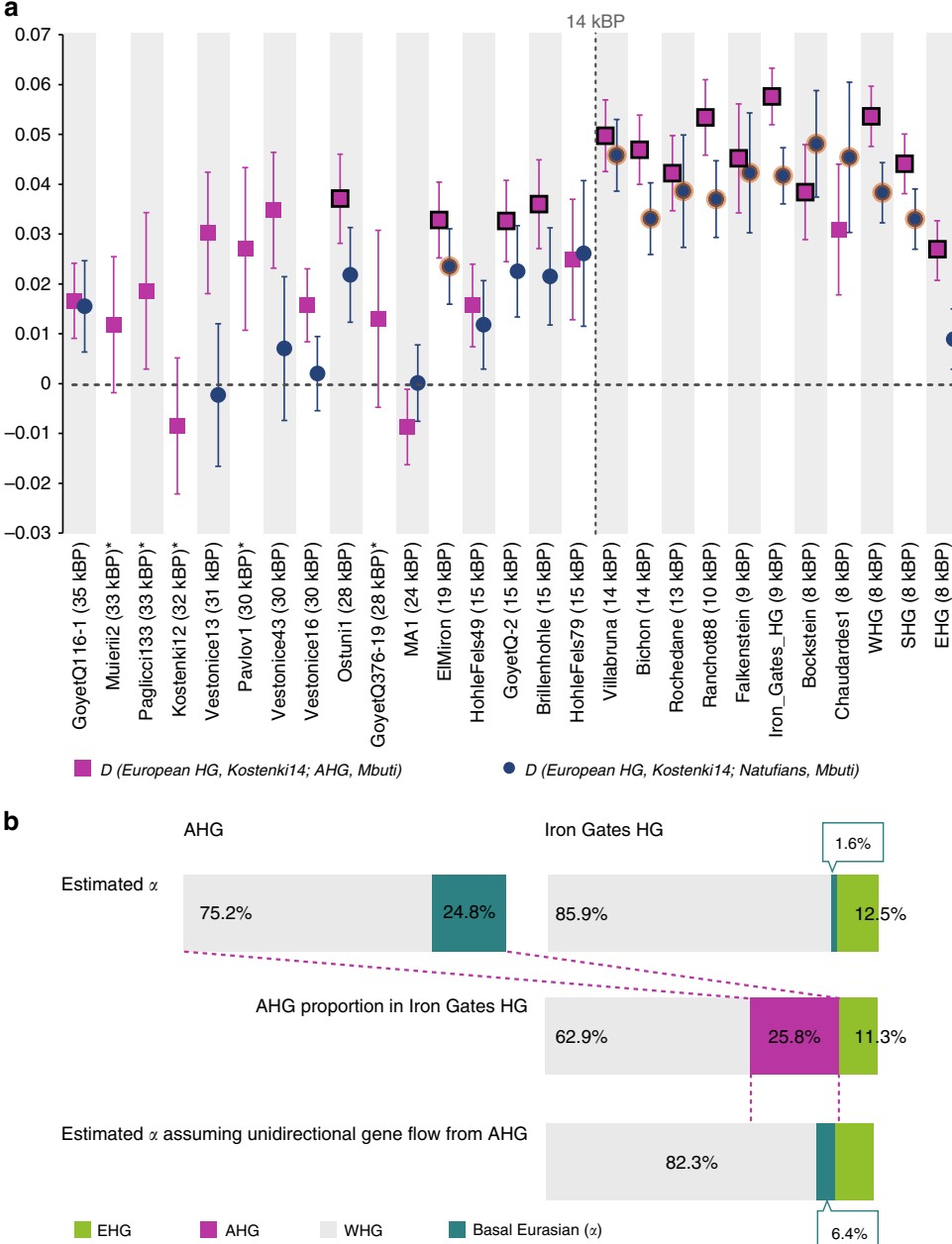

**Fig. 3** Genetic links between Near-Eastern and European hunter-gatherers. **a** Genetic affinity between Near-Eastern and European hunter-gatherers increases after 14,000 years ago as measured by the statistic $D$(European HG, Kostenki14; Natufian/AHG, Mbuti). Vertical lines mark ± 1 SE. Data points for which $D > 3$ SE are outlined. *Kostenki14* serves here as a baseline for the earlier European hunter-gatherers. Statistics including all analyzed European hunter-gatherers are listed in Supplementary Data 5. Individuals marked with an asterisk did not reach the analysis threshold of over 30,000 single-nucleotide polymorphisms (SNPs) overlapping with *Natufian/AHG*. **b** Basal Eurasian ancestry proportions ($\alpha$) as a marker for Near-Eastern gene flow. Mixture proportions inferred by qpAdm for the Anatolian hunter-gatherer (AHG) and the Iron Gates hunter-gatherers (Iron Gates HG) are schematically represented[6]. The lower schematic shows the expected $\alpha$ in Iron Gates HG under assumption of unidirectional gene flow, inferred from $\alpha$ in the AHG source population. The observed $\alpha$ for Iron Gates HG is considerably smaller than expected; thus, the unidirectional gene flow from the Near East to Europe is not sufficient to explain the affinity between Iron Gates HG and AHG. Source data are provided as a Source Data file

in 1 ml of extraction buffer (0.45 M EDTA, pH 8.0, and 0.25 mg ml$^{-1}$ proteinase K) at 37 °C for over a 12–16 h. Subsequently, DNA was bound to a silica membrane using a binding buffer containing guanidine hydrochloride and purified in combination with the High Pure Viral Nucleic Acid Large Volume Kit (Roche). DNA was eluted in 100 µl of TET (10 mM Tris-HCl, 1 mM EDTA, pH 8.0, and 0.05% Tween-20). One extraction blank was taken along. The extracts were then shipped to Jena, Germany where downstream processing was performed.

In Jena, Germany, all pre-amplification steps were performed in dedicated aDNA facilities of the Max Planck Institute for the Science of Human History (MPI-SHH). The inner ear part of the petrous bones of the individuals from Kfar

HaHoresh and Ba'ja was sampled by drilling[24] and DNA was extracted from 76 to 109 mg of the bone powder. An extraction of ~100 mg pulverized bone from the Pınarbaşı individual ZBC was done in the Jena facility in addition to the Liverpool extraction (the sequenced data from the two extracts of individual ZBC were merged in downstream analysis after passing the quality control step). All extractions followed the same protocol as cited for Liverpool. A 20 µl aliquot from each extract was used to prepare an Illumina double-stranded, double-indexed DNA library following established protocols[11,25]. Deaminated cytosines that result from DNA damage were partially removed using uracil-DNA glycosylase and endonuclease VIII, but still retained in terminal read positions as a measure of

aDNA authentication[26]. A negative library control (H$_2$O) was taken along for each experiment. Unique combinations of two indexes (8 bp length each) were assigned to each library. The indexes were then attached through a ten-cycle amplification reaction using the *Pfu Turbo Cx Hotstart DNA Polymerase* (Agilent), the PCR products purified using a Qiagen MinElute kit (Qiagen), and then eluted in TET (10 mM Tris-HCl, 1 mM EDTA, pH 8.0, and 0.05% Tween-20). Subsequently, indexed libraries were amplified using Herculase II Fusion DNA polymerase, following the manufacturer's protocol, to a total of 10$^{13}$ DNA copies per reaction and again purified using a Qiagen MinElute kit (Qiagen) and eluted in TET (10 mM Tris-HCl, 1 mM EDTA, pH 8.0 and 0.05% Tween-20). Finally, all samples were diluted and pooled (10 nM) for sequencing. The indexed amplified libraries were also used for two previously published downstream in-solution enrichments: a protocol targeting 1,237,207 genome-wide SNPs ("1240k capture"[12]) and one targeting the entire human mitochondrial genome[27].

The "1240k capture" is an established in-solution enrichment assay based on hybridization of the indexed libraries to DNA probes[12,13,27,28]. The targeted SNP panel is a combination of the two separate SNP sets first reported by Haak et al.[13] and by Fu et al.[28] and further described by Mathieson et al.[12]. For each of the ~1.2 million target SNPs, we used four distinct 52-bp-long probes: two flanking the target SNP from each side and the other two centered on the SNP matching with the reference and alternative allele, respectively[28]. The capture was performed following the published protocol described in detail in the SI text sections 3.2–3.3 of Fu et al.[28] with modified hybridization conditions of 65 °C for about 24 h.

Both the initial shotgun and target-enriched libraries were single-end sequenced on an Illumina Hiseq 4000 platform ($1 \times 75 + 8 + 8$ cycles). Sequenced reads were demultiplexed allowing one mismatch in each index and further processed using EAGER (v 1.92.54)[29]. First, adapter sequences were clipped and reads shorter than 30 bp were discarded using AdapterRemoval (v 2.2.0)[30]. Adapter-clipped reads were subsequently mapped with the BWA aln/samse programs (v 0.7.12)[31] against the UCSC genome browser's human genome reference hg19 with a lenient stringency parameter ("-n 0.01"). We retained reads with Phred-scaled mapping quality scores ≥20 and ≥30 for the whole genome and the mitochondrial genome, respectively. Duplicate reads were subsequently removed using DeDup v 0.12.2[29]. Pseudo-diploid genotypes were generated for each individual using pileupCaller, which randomly draws a high quality base (Phred-scaled base quality score ≥30) mapping to each targeted SNP position (https://github.com/stschiff/sequenceTools). To prevent false SNP calls due to retained DNA damage, two terminal positions in each read were clipped prior to genotyping. The genotyping produced between 129,406 and 917,473 covered targeted SNPs and a mean coverage ranging between 0.16 and 2.9 fold per individual (Table 1).

**Dataset.** We merged the newly reported ancient data and data reported by Mathieson et al. 2018[18] with a dataset that has been described elsewhere[6]. This dataset includes 587 published ancient genomes[6–9,12,14,17,23,32–35] and genomes from 2706 individuals, representing world-wide present-day populations[6,36] that were genotyped on the Affymetrix Axiom$^{TM}$ Genome-Wide Human Origins 1 array[4] ("HO dataset") with a total of 597,573 SNP sites in the merged dataset. To minimize bias from differences in analysis pipelines, we re-processed the raw read data deposited for previously published Neolithic Anatolian genomes[9] (labeled Tepecik_pub and Boncuklu_pub) in the same way as described for the newly reported individuals.

**aDNA authentication and quality control.** We estimated authenticity of the ancient data using multiple measures. First, blank controls were included and analyzed for extractions as well as library preparations (Supplementary Data 8). Second, we assessed levels of DNA damage in the mapped reads using mapDamage (v 2.0)[37]. Third, we estimated human DNA contamination on the mitochondrial DNA using schmutzi[38]. Last, we estimated nuclear contamination in males with ANGSD (v 0.910)[39], which utilizes haploid X chromosome markers in males by comparing mismatch rates of polymorphic sites and adjacent ones (that are likely to be monomorphic). The genetic sex of the reported individuals was determined by comparing the genomic coverage of X and Y chromosomes normalized by the autosomal average coverage. To avoid bias caused by grouping closely related individuals into a population, we calculated the pairwise mismatch rates of the Boncuklu individuals following a previously reported method[40] (Supplementary Data 9).

Five of the 12 individuals reported here were excluded from the population genetic analysis: two due to a high genomic contamination level (>5%) and three due to low amount of analyzable data (<10,000 SNPs covered); (Supplementary Data 1).

**Principal component analysis.** We used the smartpca software from the EIGENSOFT package (v 6.0.1)[41] with the lsqproject option to construct the principal components of 67 present-day west Eurasian groups and project the ancient individuals on the first two components (Supplementary Figure S8).

**ADMIXTURE analysis.** We used ADMIXTURE (v 1.3.0)[42] to perform a maximum-likelihood unsupervised clustering of 3293 ancient and present-day individuals in the HO merged dataset, allowing the number of clusters (k) to range

between 2 and 20. Pruning for linkage disequilibrium (LD) was done by randomly removing one SNP from each pair with genotype $r^2 \geq 0.2$, using PLINK (v 1.90)[43,44]; (–indep-pairwise 200 25 0.2). The analysis was replicated five times for each k value with random seeds and the highest likelihood replicate is reported (Supplementary Figures 1 and 9). Five-fold cross-validation errors were calculated for each run. Using the same settings, we additionally preformed the clustering on a smaller sample size of 983 ancient and modern west Eurasian individuals, which produced a clustering pattern comparable to that of the larger dataset.

**D-statistics.** To estimate allele frequency correlations between populations, *D*-statistics were computed using the *qpDstat* program (v 701) of the ADMIXTOOLS package[45] (v 4.1) with default parameters. *D*-statistics provide a robust and sensitive test of gene flow and are preferable for low quantity data analysis (typical of Archeogenetic studies) as they are insensitive to post-admixture drift, including artifactual drift due to a limited sample size[45]. In order to determine whether a test population is symmetrically related to populations X and Y, the *D*-statistic *D* (*X, Y; Test, Outgroup*) was used. In particular, when comparing the affinity of different European hunter-gatherers to Near-Eastern ones in the *D*-statistic of the form *D* (*European HG1, European HG2; Near Eastern HG, Outgroup*), both the central African *Mbuti* and the Altai Neanderthal (*Altai_published.DG*) were used to check if the differing level of Neanderthal ancestry in these hunter-gatherers affects the results. Otherwise, Mbuti was used as the single outgroup. The above statistics are reported when more than 30,000 SNP positions were overlapping between the four analyzed populations. To further validate the *D*-statistics of the form *D* (*Anatolian 1, Anatolian2; test, Mbuti*) beyond the jackknifing performed by *qpDstat*, we compared the inferred *D*-statistics based on the population mean to the distribution of the *D*-statistic when individuals are permuted between populations. We performed the permutation tests in the following settings: (1) for the *D*-statistics of the form *D* (*AAF\*, AHG\*; test, Mbuti*), we performed all five possible permutations. In each permutation, we placed one out of the five AAF individuals into the second position (*AHG\**) while placing the other four individuals and the AHG individual into the first position (*AAF\**) (Supplementary Data 11). To obtain the distribution within AAF we repeated the analysis, but now excluding AHG. The same set of global modern and ancient populations as in the original test was used as the "test." (2) For the *D*-statistics of the form *D* (*ACF\*, AAF\*; test, Mbuti*) a total of 1,000 permutations were performed, in addition to the original test, for each of the four "test" populations that had the most positive values in the original observed statistic (i.e., *Levant_N, Natufian, Greece_EN, Balkans_Neolithic*). In each test, we randomly chose five out of 30 individuals (5 AAF and 25 ACF) and placed them into the second position (*AAF\**) while placing the rest into the first position (*ACF\**). Empirical *P*-values were calculated by dividing the number of permutations with a *D*-statistic equal to or greater than the original observation by the total number of permutations (i.e., 1001).

**Modeling ancestry proportions.** We used the qpWave (v400) and qpAdm (v 632) programs of ADMIXTOOLS[6,13] to test and model admixture proportions in a studied population from potential source populations (reference populations). As the explicit phylogeny is unknown, a diverse set of outgroup populations (Supplementary Notes2–4) was used to distinguish the ancestry of the reference populations.

For estimating admixture proportions in the tested populations, we used a basic set of seven outgroups including present-day populations (Han, Onge, Mbuti, Mala, Mixe) that represent a global genetic variation and published ancient populations such as Natufian[6], which represents a Levantine gene pool outside of modern genetic variation and the European Upper Paleolithic individual Kostenki14[17]. As a prerequisite for the admixture modeling of the target population, we tested whether the corresponding set of reference populations can be distinguished by the chosen outgroups using qpWave[6] (Supplementary Note 3). In some cases, when a reference population did not significantly contribute to the target in the attempted admixture models, it was removed from the reference set and added to the basic outgroup set in order to increase the power to distinguish the references. In cases where "Natufian" was used as a reference population, we instead used the present-day Near-Eastern population "BedouinB" as an outgroup.

For estimations of Basal Eurasian ancestry, we followed a previously described qpAdm approach[6] that does not require a proper proxy for the Basal Eurasian ancestry, which is currently not available in unadmixed form. This framework relies on the basal phylogenetic position of both Basal Eurasian and an African reference (the ancient Ethiopian *Mota* genome[19]) relative to other non-Africans. Thus, by using a set of outgroups that includes eastern non-African populations (Han; Onge; Papuan) and Upper Paleolithic Eurasian genomes (Ust'-Ishim[46], Kostenki14, MA-1[47]), but neither west Eurasians with detectable basal Eurasian ancestry nor Africans, the mixture proportion computed for *Mota* (α) can be used indirectly to estimate the Basal Eurasian mixture proportion of west Eurasian populations.

**Mitochondrial DNA analysis.** The endogenous mitochondrial consensus sequences were inferred from the output of schmutzi[38], using its log2fasta program and a quality cutoff of 10. Mitochondrial haplotypes were established by aligning these consensuses to rCRS[48] using the online tool haplosearch[49]. The coverage of

each of the reported SNPs was confirmed by visually inspecting the bam pileup in Geneious (v11.0.4)[50]. The resulting consensus sequences were then analyzed with HaploFind[51] and Haplogrep[52] to assign mitochondrial haplogroups and double-checked with the rCRS oriented version of Phylotree[53].

**Y-chromosome analysis.** To assign Y-chromosome haplogroups we used yHaplo[54]. Each male individual was genotyped at 13,508 ISOGG consortium SNP positions (strand-ambiguous SNPs were excluded) by randomly drawing a single base mapping to the SNP position, using the same quality filters as for the HO dataset. In addition to the yHaplo automated haplogroup designations, we manually verified the presence of derived alleles supporting the haplogroup assignment.

**Phenotypic traits analyses.** We tested for the presence of alleles related to biological traits that could be of interest in the geographical and temporal context of the reported ancient populations, including lactase persistence[55,56], Malaria resistance[57,58], glucose-6-phosphate dehydrogenase deficiency[59,60], and skin pigmentation[23,61,62]. The allele distribution for the SNP positions listed in Supplementary Data 7 was tabulated for each individual using Samtools mpileup (v 1.3).

**Carbon dating.** The phalanx bone from individual ZBC (Pinarbaşı) and the petrous bone from individual KFH2 (Kfar HaHoresh) were each sampled and directly radiocarbon dated at the CEZ Archaeometry gGmbH, Mannheim, Germany (Supplementary Table 1). Collagen was extracted from the bone samples, purified by ultrafiltration (fraction >30kDa), freeze-dried, and combusted to $CO_2$ in an elemental analyzer. $CO_2$ was converted catalytically to graphite. The dating was performed using the MICADAS-AMS of the Klaus-Tschira-Archäometrie-Zentrum. The resulting $^{14}C$ ages were normalized to $d^{13}C = -25\%$[63] and calibrated using the dataset INTCAL13[64] and the software SwissCal 1.0[65].

**Reporting summary.** Further information on experimental design is available in the Nature Research Reporting Summary linked to this article.

## Data availability

All Alignment data (BAM) produced in this study is deposited in the European Nucleotide Archive (ENA) under the accession numbers (Study PRJEB24794). Other data supporting the findings of the study are available in this article and its Supplementary Information files, or from the corresponding authors upon request. The human skeletal specimens from Pınarbaşı are housed in the Karaman Museum, Turkey. They are available for study with the authorization of Professor Douglas Baird and the Directorate of Antiquities and Museums of Turkey. The human skeletal specimens from Boncuklu are housed in the Boncuklu excavation depot, under the purview of Konya Museum, Turkey. They are available for study with the authorization of Professor Douglas Baird and the Directorate of Antiquities and Museums of Turkey. The skeletal specimens of the Kfar Hahoresh individual are housed in the anthropological collection, the Dan David Center for Human Evolution and Biohistory Reasearch, the Sackler faculty of medicine, Tel Aviv University, Israel and are available upon request and authorization of Prof. Nigel Goring-Morris, Prof. Israel Hershkovitz, and Dr. Hila May. The skeletal specimens of the Baj'a individual are housed in the Department of Anthropology, German Archeological Institute, Berlin, Germany and are available upon request and authorization of Dr. Julia Gresky.

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

## Acknowledgements

We thank G. Brandt, A. Wissgott, F. Aron, M. Burri, C. Freund, and R. Stahl (MPI-SHH) for their support in laboratory work, M. Oreilly for graphic support, A. Gibson for help in proofreading, and the members of the population genetics group in the Department of Archaeogenetics, SHH-MPI for their input and support. This work was supported by the Max Planck Society. Experimental work at LJMU and LR stay at the MPI-SHH were funded with an internal grant from LJMU Faculty of Science to EFD. The funding for the Ba'ja project was granted by the German Research Foundation (GZ: 80 1599/14–1) and ex oriente e.V., Berlin. Kfar HaHoresh fieldwork was supported (to N.G.-M.) by the Israel Science Foundation (Grants 840/01, 558/04, 755/07, 1161/10), The National Geographic Society (Grant 8625/09), and The Irene Levi Sala CARE Foundation. The Konya Plain fieldwork was funded by The British Institute in Ankara, British Academy (Research Development Award BR100077), a British Academy Large Research Grant LRG 35439, Australian Research Council (grants DP0663385 and DP120100969), National Geographic award GEFNE 1-11, University of Oxford (Wainwright Fund), Australian Institute for Nuclear Science and Engineering (AINSE awards AINGRA05051 and AINGRA10069), Wenner-Gren Foundation for Anthropological Research (Postdoctoral Research Grant 2008 The Origins Of Farming In The Konya Plain, Central Anatolia), Institute for Field Research.

## Author contributions

J.K., E.F.-D., I.H., C.J., W.H. and M.F. conceived the study. D.B., A.F., G.M., J.P., I.H., H. M., N.G.M., M.B. and J.G. provided archeological material. D.B., J.P., A.F., G.M., I.H., H. M., N.G.-M., M.B., J.G. and P.W.S advised on the archeological background and interpretation. D.B., J.P., E.F.-D., N.G.-M., M.B. and J.G. wrote the archeological and sample background sections. M.F., L.R. and R.A.B. performed the laboratory work. M.F., E.F.-D., L.R., C.P. and C.J. performed data analysis. M.F., E.F.-D., C.J. and J.K. wrote the manuscript with input from all coauthors.

## Additional information

**Competing interests:** The authors declare no competing interests.

