## [Peer Review File · Nature Communications]

Reviewers' comments:

Reviewer #1 (Remarks to the Author):

This is a report of new data from a challenging context - human remains from Anatolia. It includes a 15kya hunter gatherer, PPN farmers and PN farmers. The group has worked hard to retrieve data from samples with poor preservation and the data seems credible. There is a conclusion of continuity in the region and some suggestion of different layers of admixture with neighbouring, divergent genome types. I think this contains publishable inference.

Some criticisms:

- A comment on methods. This paper describes the retrieval of information from very challenging samples. However, there is nothing of depth in the methods section that might help other researchers planning such work. This contrasts, for example with Fregel et al, PNAS recently, a study of similar magnitude, which carries a considered discussion about capture methods in its supplement. The primary reference for the capture technique quoted has no detail either. There is a proprietary problem with obtaining the capture array from the same source as this group but that does not excuse the lack of description. I think as a minimum the supplement should describe the type of capture (type of anchors, probe length, how many baits per SNP, are there baits complementary to each alternate allele, DNA/RNA). Also reaction conditions - amount of target DNA, concentration of baits used, reaction solution, annealing temperature and time, purifications methods, cycles used etc.) This is normal scientific publishing practice and certainly a minimum for a journal such as this. Capture data may show some biases compared to shotgun data and its basis needs to be clear.

- the inference and analysis falls within the standard approaches in the field. It may change with addition of more data - small sample numbers is an inevitable feature of aDNA research in challenging contexts. One aspect that requires more explanation is the section discussing Basal European ancestry (alpha) lines 173-191. This is used as if it is an accepted, well known concept, which may not be true. I think it needs some introduction and justification.

Reviewer #2 (Remarks to the Author):

In this paper Feldman and colleagues are able to sequence the genome of a Paleolithic Anatolian as well as add to the existing Anatolian and Levantine Neolithic genomes that are currently published. The former in particular is important, as there has been a gap in our knowledge of the origins of Anatolian Neolithic component that becomes so prominent in Neolithic Europe, given that the region is intermediate of the Near East where there appears to be the adoption of agriculture from the existing hunter-gatherer population. This new AHG sample begins to answer the question, showing that the Paleolithic Anatolians looked a lot like the Neolithic Anatolians whose descendants would then spread through Europe, thus finally linking the initial cultural adoption we observe in the Near East with the later demic diffusion we see across Europe. In addition, the authors bring some clarification to the apparent hunter-gatherer connections between Late Pleistocene Europe and the Near East. Though many of the actual migration processes are not yet fully fleshed out due to the sparsity of the data temporally and geographically (as the authors appropriately describe), this paper provides an excellent foundation and will be important for further ancient DNA that might be recovered from the region in the future. It will be of interest to a wide audience (genetics, archaeologists) who are interested in the Paleolithic/Neolithic transition in Europe from the Near East.

By established metrics the ancient SNP enrichment data seem reliable in terms of processing and contamination, and the analysis and results, though using fairly standard methods, are robust in their interpretation and appropriately applied. The language gets a little bit sloppy in the discussion when describing different changes in ancestry, but otherwise the paper is well written. The only major addition I would like to see would be perhaps a cartoon figure at the end that shows the different migrations/mixture scenarios identified in the paper, as otherwise it can be a bit hard to see the main points through the in depth description of the D-statistics.

I also have the following minor comments:

Line 77: "By analyzing this data"

Probably don't need this clarification.

Line 78: "thus represent a previously undescribed population."

Given they look a lot like other Anatolian Neolithic and European Neolithic samples, this does not seem like a true statement (though it is the earliest representation of this population).

Line 101: "with a slight leftward shift".

If there a way to assess the error associated with this plotting (i.e. the significance of a "leftward shift". Though D-statistics show some differences, it is not clear that this is actually the same signal of the PCA (which may not have the resolution for this inference) given there is a natural spread to all Neolithic Anatolians and Europeans in the PCA.

Line 103: "are positioned between Mesolithic western European hunter-gatherers (WHG) at the far left and Levantine Epipalaeolithic Natufians at the far right"

I think it is more appropriate to speak of where they lie relative to each on PC1 and PC2 (i.e. intermediate of the extremes of PC1, but overlapping eigenvalues on PC2) rather than some arbitrary "left" and "right" designation.

Line 105: "The newly reported Neolithic farmers (BAJ001 and KFH2) are positioned near the published ones (Supplementary Text S2)."

Please mention you are talking about the published LEVANT Neolithic farmers for readability and keeping track.

Line 106: "In ADMIXTURE analysis, AHG, AAF and ACF are all modeled as a mixture of two components that are each maximized in Natufians and WHG, consistent with their positions in PCA (fig. S1)."

The ADMIXTURE results show them to be much more weighed towards the green Natufian component. The WHG component appears quite small, versus the intermediate position in the PCA. This statement needs a little rewording as the results are not so congruent as described.

Line 123: "AAF is slightly shifted upwards compared to AHG in the PCA"

Again, I think it is better/more accurate to describe in terms of PCs, not "upwards" or "downwards".

Line 124: "when compared to AHG by $D(\text{AAF}, \text{AHG}; \text{test}, \text{Mbuti})$, the AAF early farmers show extra affinity with early Holocene populations from Iran or Caucasus and with present-day South Asians, who have also been genetically linked with Iranian/Caucasus ancestry"

Given AHG is just one pseudo-diploid genome, to what extent are these results significant? I do not see Z-scores in Fig 2 or S2?

Line 137: "In contrast, we find that the later ACF individuals share more alleles with the early Holocene Levantines than AAF do, as shown by positive $D(\text{ACF}, \text{AAF}; \text{Natufian/Levant_N}, \text{Mbuti}) \geq$

3.84 SE (Fig. 2B, fig. S3 and data table S3). Ancient Iran/Caucasus populations and contemporary South Asians do not share more alleles with ACF ($|D| < 3.3$ SE)."

Why is $|3.3|$ not considered significant, but $|3.84|$ is? Seems a bit arbitrary. Is D(Natufian/Levant, Iran/Caucasus, ACF) significant?

Line 159: "As is the case for present-day Near-Easterners, the Near-Eastern hunter gatherers share more alleles with the Later European HG than with the Earlier European HG, shown by the significantly positive statistic D(Later European HG, Earlier European HG; AHG/Natufian, Mbuti) (Fig. 3A and data table S5)"

It would be easier to transition to understanding the figure more quickly if you mention in the main text that Kostenki14 is your early HG representative. Could significance be indicated somewhere on the figure? Otherwise the 14k is a bit arbitrary, especially with overlapping SEs on each side of the time point.

Fig 3 maybe needs a little more clarification on the figure itself, it took me a while to figure out what it was showing (e.g. that the text "Model assuming...." Applied to both the last and second to last bar.

Line 192: "The uniparental marker analysis..."

I am really not sure what this section adds to the paper. It is not discussed at all in the discussion as far as I can tell.

Line 220: "Interestingly, while the local population structure remains highly stable,"

I do not think such a strong statement can be made given there is only one AHG sample, which cannot tell you about local population structure per se (you would need genomes for multiple individuals [at least two] to get some estimate of local population structure at any particular time point). I would suggest to tone down or rephrase.

Line 222: "External genetic contributions, associated with two distinct early farming populations of the Fertile Crescent, substituted about 10% and 20% of indigenous ancestry each"

Clumsy and ambiguous statement. How is ancestry "substituted"? Can we not be more precise in the description, and use population genetics terminology?

Line 228: "how these gene flows were introduced"

Use of "gene flow" in this section sounds weird. Can you rephrase?

Reviewer #3 (Remarks to the Author):

While these new data allow some important observations about genetic relatedness among Anatolian Hunter-Gatherers (AHG) and pre-Ceramic (AAF) and post-Ceramic (ACF) Anatolian Farmers, I think the strength of evidence behind observations involving possible outside introgression is lacking or difficult to assess. In particular it is unclear whether many of the observed signals are explained by admixture among genetically different groups, or whether signals are consistent with models that do not have any explicit admixture events. For example, you conclude that qpAdm results suggest AHG "results support a late Pleistocene presence of both ancestries in a mixed form", which suggests admixture, but presumably it is also consistent with AHG being equally genetically similar to the included Neolithic Levantine versus WHG samples, without any admixture between them? Notably the inferred qpADM proportions for AHG are very different than what might be interpreted by your ADMIXTURE plot (Fig S1); is there a reason this is the case? In general it is not clear how strongly you can conclude admixture based on qpAdm analyses. Instead did you try analyses like f3-statistics that

explicitly test for admixture, rather than f_4 tests that assesses a more general "genetic affinity"? Or does qpAdm already include such f_3 -statistics when it concludes a group (e.g. AHG, AAF, ACF) is admixed? If so, can you include the f_3 tests that explicitly test for admixture? My understanding is that qpAdm is similar to TREEMIX, which does not use f_3 -statistics and often in practice infers admixture when it is not present and/or misses admixture that is present. In other words, how do we know the values in Tables S4-S9 are not reflecting a more general recent shared ancestry without involving admixture? It would be good to clarify these points -- i.e. what can be interpreted reliably from qpAdm inference.

As another example, one of the major points of the article is that there is genetic continuity between Anatolian Hunter Gatherers (AHG) and pre-Ceramic Anatolian Farmers (AAF). Your PCA (Fig 1) and ADMIXTURE (Fig S1) plots are consistent with this. However you also hint that there appears to be admixture (or "incoming ancestry") related to an Iranian Neolithic-like source in the AAF, which I think is less clear. While it is true that your plots (Fig 2A) and Fig S2 show a greater affinity between AAF and populations to the east relative to AHG, it is not clear this is due to admixture. First off, these admixture signals again seem inconsistent with your ADMIXTURE figure (Fig S1), which does not suggest that AHG and AAF are different (though maybe hinting that 1-2 AAF samples have very slightly increased Iran_N-like DNA). As a hypothetical example, if AAF were directly continuous from AHG without any outside ancestry contributions, could you get qpAdm values of inferred Iran_N ancestry similar to that you see in Table S6 if e.g. your AHG sample happened to come from a bottlenecked population that thus does not perfectly reflect the AHG population from which AAF derives? You also only have one AHG individual, which makes these analyses complicated and presumably prone to a lack of power, as suggested by your long confidence intervals when jack-knifing. A discussion and/or test of this uncertainty would be helpful.

For example, when testing for additional ancestry in AAF that is lacking in AHG, your null is that AHG and AAF genetically are a single population. So could you do a permutation test where you swap your single AHG with a single AAF sample and re-calculate $D(\text{AAF}^*, \text{AHG}^*; \text{test}, \text{Mbuti})$, where AHG^* is this single AAF sample and AAF^* now includes all other AAF plus the AHG (or perhaps simply excluding the AHG)? Try replacing AHG^* with each of the AAF samples and note how often you get patterns similar to that in Fig 2A/Fig S2 (e.g. how often your maximum D value is greater than the max in Fig 2A, and/or how often this D score is positive when "test" comprises one of the eastern populations with positive D in Fig 2A). This would assess whether there really is a tendency for AHG to be less related to the eastern test groups, or whether by chance a single AAF individual might also have less affinity to these eastern groups relative to the other AAFs. Indeed your Supp data table S10 suggests the latter will be the case? In general it would help to run such tests, in addition to f_3 statistics to test explicitly for admixture, to get an idea of certainty about these claims. It could be that only some AAF carry this N.Iran-like ancestry (i.e. as indicated in Fig S1) and/or that these signals can be explained by chance ancestry differences in a homogeneous population rather than genuine admixture. As mentioned above, could they also reflect a lack of perfect surrogates?

You can perhaps do a similar permutation procedure to assess the reliability of claims about ACF having more ancient Levant-like ancestry relative to AAF -- e.g. what happens to Fig 2B and Fig S3 if you randomly permute AAF/ACF labels; do the affinities in Fig 2B and Fig S3 converge to 0 as expected?

Figure 3 is difficult to follow. First off, it would be helpful to re-acquaint the reader with the idea of Basal Eurasian ancestry, including tree plots possibly, and the intuition (again perhaps using figures in the SOM) behind the test you are doing here to infer the Basal Eurasian component (i.e. alpha). Again uncertainty is difficult to assess here. In particular it seems hard to believe that estimating ~0% ancestry in Iron Gates HS is inconsistent with the truth being ~6% -- are estimates really that precise,

even though you're using only a single AHG sample? Some discussion on this, or some testing using simulations, would be helpful, because it is difficult to understand how strong a statement can be made here.

minor comments

-- throughout you report ancestry percentages to the 10th decimal place -- can it really be this precise?

Fig S1 -- there appears to be little difference between $K=6-15$ here, presumably in part because you are running ADMIXTURE on a world-wide collection of samples (Fig S5), most of which are not relevant for what you are studying here. Can you run ADMIXTURE on only samples relevant to this analysis, e.g. excluding Africa and other regions? This might increase power to assess whether there are genetic differences between AAF, ACF, AHG, etc, and how these groups relate to ancient Levantines, Iranian Neolithics, etc.

SOM references -- The numbers are higher (e.g. 56, 57) than what's listed in these sections, so need to be re-set presumably?

Reviewers' comments:

Reviewer #1 (Remarks to the Author):

This is a report of new data from a challenging context - human remains from Anatolia. It includes a 15kya hunter gatherer, PPN farmers and PN farmers. The group has worked hard to retrieve data from samples with poor preservation and the data seems credible. There is a conclusion of continuity in the region and some suggestion of different layers of admixture with neighbouring, divergent genome types. I think this contains publishable inference.

We appreciate the positive opinion of the reviewer on our work.

Some criticisms:

- A comment on methods. This paper describes the retrieval of information from very challenging samples. However, there is nothing of depth in the methods section that might help other researchers planning such work. This contrasts, for example with Fregel et al, PNAS recently, a study of similar magnitude, which carries a considered discussion about capture methods in its supplement. The primary reference for the capture technique quoted has no detail either. There is a proprietary problem with obtaining the capture array from the same source as this group but that does not excuse the lack of description. I think as a minimum the supplement should describe the type of capture (type of anchors, probe length, how many baits per SNP, are there baits complementary to each alternate allele, DNA/RNA). Also reaction conditions - amount of target DNA, concentration of baits used, reaction solution, annealing temperature and time, purification methods, cycles used etc.) This is normal scientific publishing practice and certainly a minimum for a journal such as this. Capture data may show some biases compared to shotgun data and its basis needs to be clear.

We agree with the reviewer that it is important to provide details of the experimental methods used in this study. In the revised manuscript, we now provide a section (p. 13, lines 296-303) that provides a clear overview of probe design and reaction conditions, referencing the relevant studies which originally reported the assay. We would like to highlight that the '1240K capture' used for targeted DNA enrichment in this study has been well established and described in detail in previous publications (Fu et al. 2013 PNAS 110:2223; Fu et al. 2015 Nature 524:216; Haak et al. 2015 Nature 522:207; Mathieson et al. 2015 Nature 528:499). It has been also frequently used in ancient DNA studies for the last three years (e.g. Harney et al. 2018 Nat Commun 9:3336; Lazaridis et al. 2017 Nature 548:214; Lazaridis et al. 2016 Nature 536:419; Mathieson et al. 2018 Nature 555:197; Fu et al. 2016 Nature 536:419).

- the inference and analysis falls within the standard approaches in the field. It may change with addition of more data - small sample numbers is an inevitable feature of aDNA research in challenging contexts. One aspect that requires more explanation is the section discussing Basal European ancestry (alpha) lines 173-191. This is used as if it is an accepted, well known concept, which may not be true. I think it needs some introduction and justification.

The concept of "Basal Eurasian ancestry" is indeed a relatively new finding referring to a hypothetical lineage that is a common outgroup of Paleolithic/Mesolithic European hunter-gatherers and ancient/present-day eastern Eurasians. The idea of Basal Eurasian ancestry was first introduced by Lazaridis et al (2014, Nature 513:409) to explain an asymmetric relationship of eastern Eurasians to Mesolithic European hunter-gatherers and to early Neolithic European farmers. In the revised manuscript, we now clarify the concept of Basal Eurasian ancestry and justify our logic to use it in disentangling the direction of gene flow between Europe and the Near East in the Results (pp. 8-9, lines 196-202) as well as in the Introduction (p. 3, lines 66-74) of the main text. We also provide a more detailed description on how we model the Iron Gates HG to test the one way gene flow scenario from the Near East (p. 9, lines 203-207).

Reviewer #2 (Remarks to the Author):

In this paper Feldman and colleagues are able to sequence the genome of a Paleolithic Anatolian as well as add to the existing Anatolian and Levantine Neolithic genomes that are currently published. The former in particular is important, as there has been a gap in our knowledge of the origins of Anatolian Neolithic component that becomes so prominent in Neolithic Europe, given that the region is intermediate of the Near East where there appears to be the adoption of agriculture from the existing hunter-gatherer population. This new AHG sample begins to answer the question, showing that the Paleolithic Anatolians looked a lot like the Neolithic Anatolians who's descendants would then spread through Europe, thus finally linking the initial cultural adoption we observe in the Near East with the later demic diffusion we see across Europe. In addition, the authors bring some clarification to the apparent hunter-gatherer connections between Late Pleistocene Europe and the Near East.

Though many of the actual migration processes are not yet fully fleshed out due to the sparsity of the data temporally and geographically (as the authors appropriately describe), this paper provides an excellent foundation and will be important for further ancient DNA that might be recovered from the region in the future. It will be of interest to a wide audience (genetics, archaeologists) who are interested in the Paleolithic/Neolithic transition in Europe from the Near East.

We appreciate the positive opinion of the reviewer on our study.

By established metrics the ancient SNP enrichment data seem reliable in terms of processing and contamination, and the analysis and results, though using fairly standard methods, are robust in their interpretation and appropriately applied. The language gets a little bit sloppy in the discussion when describing different changes in ancestry, but otherwise the paper is well written. The only major addition I would like to see would be perhaps a cartoon figure at the end that shows the different migrations/mixture scenarios identified in the paper, as otherwise it can be a bit hard to see the main points through the in depth description of the D-statistics.

In the revised manuscript, we now provide a map-based inset for Figs. 2A and 2B to visualize the gene flows that we detect using D-statistics.

I also have the following minor comments:

Line 77: "By analyzing this data"
Probably don't need this clarification.

We have deleted this sentence from the manuscript.

Line 78: "thus represent a previously undescribed population."

Given they look a lot like other Anatolian Neolithic and European Neolithic samples, this does not seem like a true statement (though it is the earliest representation of this population).

Following the reviewer's suggestion, we have removed this statement from the manuscript.

Line 101: "with a slight leftward shift".

If there a way to assess the error associated with this plotting (i.e. the significance of a "leftward shift". Though D-statistics show some differences, it is not clear that this is actually the same signal of the PCA (which may not have the resolution for this inference) given there is a natural spread to all Neolithic Anatolians and Europeans in the PCA.

PCA and ADMIXTURE are powerful and popular methods to summarize the genetic structure within data, but they are descriptive methods rather than a platform for formal statistical testing of admixture. Accordingly, we used them to provide a visual summary and to generate hypotheses of population relationship to be further investigated by formal statistical tests, such as D-statistics and qpAdm. In the revised manuscript, we modified our text to highlight that we relied on formal statistical frameworks to test the significance of the descriptive patterns observed in PCA and ADMIXTURE analyses (p. 5, lines 117-119).

Line 103: “are positioned between Mesolithic western European hunter-gatherers (WHG) at the far left and Levantine Epipalaeolithic Natufians at the far right”

I think it is more appropriate to speak of where they lie relative to each on PC1 and PC2 (i.e. intermediate of the extremes of PC1, but overlapping eigenvalues on PC2) rather than some arbitrary “left” and “right” designation.

The text was changed accordingly to describe the position along PC1 or PC2 rather than left/right/up/down.

Line 105: “The newly reported Neolithic farmers (BAJ001 and KFH2) are positioned near the published ones (Supplementary Text S2).”

Please mention you are talking about the published LEVANT Neolithic farmers for readability and keeping track.

We modified the text following the reviewer’s suggestion.

Line 106: “In ADMIXTURE analysis, AHG, AAF and ACF are all modeled as a mixture of two components that are each maximized in Natufians and WHG, consistent with their positions in PCA (fig. S1).”

The ADMIXTURE results show them to be much more weighed towards the green Natufian component. The WHG component appears quite small, versus the intermediate position in the PCA. This statement needs a little rewording as the results are not so congruent as described.

Please see our previous reply regarding line 101.

Line 123: “AAF is slightly shifted upwards compared to AHG in the PCA”

Again, I think it is better/more accurate to describe in terms of PCs, not “upwards” or “downwards”.

We modified the text following the reviewer’s suggestion.

Line 124: “when compared to AHG by $D(\text{AAF}, \text{AHG}; \text{test}, \text{Mbuti})$, the AAF early farmers show extra affinity with early Holocene populations from Iran or Caucasus and with present-day South Asians, who have also been genetically linked with Iranian/Caucasus ancestry”

Given AHG is just one pseudo-diploid genome, to what extent are these results significant? I do not see Z-scores in Fig 2 or S2?

In our initial submission, we presented in Data Table 3 a whole list of $D(\text{AAF}, \text{AHG}; \text{test}, \text{Mbuti})$ statistics with the associated standard error (SE) and Z scores. Also, we provided a range of ± 1 SE in the bar plots of figures 2 and S2. In the revised manuscript, we make this information more explicitly available. First, we modified the bar plots in figure 2 to show both ± 1 and ± 3 SE (=Z score) ranges. Second, we added to the text the Z scores of the top signals (CHG and Vishwabrahmin; p. 6, lines 138-139). Third, we provide the p-value for a model comparison between models with and without the second ancestry from Neolithic Iranians (p. 6, lines 142-144). Although AHG is indeed one pseudo-diploid genome, it has a reasonable coverage (2.9x mean coverage for the target sites and 79% of target sites are covered by at least one read), comparable to the better published ancient genomes.

Line 137: “In contrast, we find that the later ACF individuals share more alleles with the early Holocene Levantines than AAF do, as shown by positive $D(\text{ACF}, \text{AAF}; \text{Natufian/Levant_N}, \text{Mbuti}) \geq 3.84$ SE (Fig. 2B, fig. S3 and data table S3). Ancient Iran/Caucasus populations and contemporary South Asians do not share more alleles with ACF ($|D| < 3.3$ SE).”

Why is $|3.3|$ not considered significant, but $|3.84|$ is? Seems a bit arbitrary. Is $D(\text{Natufian/Levant, Iran/Caucasus}, \text{ACF})$ significant?

Indeed “ $|D| < 3.3$ SE” is a typing mistake. The correct value, “ $|D| < 1.3$ SE”, was reported in data table S3 and in the Supplementary text section S3 of the initial submission. We thank the reviewer to drawing our attention to this error which is now fixed in the main text.

Line 159: “As is the case for present-day Near-Easterners, the Near-Eastern hunter gatherers share more alleles with the Later European HG than with the Earlier European HG, shown by the significantly positive statistic $D(\text{Later European HG, Earlier European HG; AHG/Natufian, Mbuti})$ (Fig. 3A and data table S5)”

It would be easier to transition to understanding the figure more quickly if you mention in the main text that Kostenki14 is your early HG representative. Could significance be indicated somewhere on the figure? Otherwise the 14k is a bit arbitrary, especially with overlapping SEs on each side of the time point.

Following the reviewer’s suggestion, we modified the main text to clarify that we used Kostenki14 as the early HG representative (p.8, lines 173-175).

In addition to the error bars (± 1 standard errors) we provided for Fig. 3A in the initial submission, we now mark data points with strong statistical support for additional allele sharing ($Z > 3$).

We acknowledge that 14,000 BP may not be a robust distinction but we would like to keep the line for a few reasons. First, for populations younger than 14,000 BP, 20 out of 22 tests are highly significant ($D > 3$ SE; 9 out of 11 populations are > 3 SE for both AHG and Natufian) while none of 16 populations older than 14,000 BP show such robust signal for both AHG and Natufian (only 5 out of 27 tests shown have $D > 3$ SE). Therefore, we think that the line indeed captures the beginning of the robust signal quite well. Second, the line was initially proposed by Fu et al. (2016 Nature 534:200) and we would like to make a cross comparison with our study easier.

Fig 3 maybe needs a little more clarification on the figure itself, it took me a while to figure out what it was showing (e.g. that the text “Model assuming...” Applied to both the last and second to last bar.

We modified Fig. 3B and its legend to clarify which bar graph belongs to which admixture model.

Line 192: “The uniparental marker analysis...”

I am really not sure what this section adds to the paper. It is not discussed at all in the discussion as far as I can tell.

We did not discuss the uniparental marker results in detail because we focused our analysis on genome-wide data, which capture probabilistic nature of demographic processes far better than a single locus. However, we believe that uniparental data are of interest and value to many research groups and while not extensively discussed in this paper are still reported by us.

Line 220: “Interestingly, while the local population structure remains highly stable,”

I do not think such a strong statement can be made given there is only one AHG sample, which cannot tell you about local population structure per se (you would need genomes for multiple individuals [at least two] to get some estimate of local population structure at any particular time point). I would suggest to tone down or rephrase.

We rephrased the main text to “we observe a continued presence of the AHG-related gene pool throughout the studied period” to clarify our message that genomes of AHG, AAF and ACF share a similar genetic profile.

Line 222: “External genetic contributions, associated with two distinct early farming populations of the Fertile Crescent, substituted about 10% and 20% of indigenous ancestry each”

Clumsy and ambiguous statement. How is ancestry “substituted”? Can we not be more precise in the description, and use population genetics terminology?

We modified the main text as follows: “In addition to the local genetic contribution from earlier Anatolian populations, Anatolian Aceramic farmers inherit about 10% of their genes from a gene pool related to the Neolithic Iran/Caucasus while later Anatolian Ceramic farmers derive about 20% of their genes from another distinct gene pool related to the Neolithic Levant” (p.11, lines 242-248).

Line 228: “how these gene flows were introduced”

Use of “gene flow” in this section sounds weird. Can you rephrase?

We rephrased the sentence to “how these genetic changes happened in Central Anatolia”.

Reviewer #3 (Remarks to the Author):

While these new data allow some important observations about genetic relatedness among Anatolian Hunter-Gatherers (AHG) and pre-Ceramic (AAF) and post-Ceramic (ACF) Anatolian Farmers, I think the strength of evidence behind observations involving possible outside introgression is lacking or difficult to assess. In particular it is unclear whether many of the observed signals are explained by admixture among genetically different groups, or whether signals are consistent with models that do not have any explicit admixture events. For example, you conclude that qpAdm results suggest AHG "results support a late Pleistocene presence of both ancestries in a mixed form", which suggests admixture, but presumably it is also consistent with AHG being equally genetically similar to the included Neolithic Levantine versus WHG samples, without any admixture between them? Notably the inferred qpADM proportions for AHG are very different than what might be interpreted by your ADMIXTURE plot (Fig S1); is there a reason this is the case? In general it is not clear how strongly you can conclude admixture based on qpAdm analyses. Instead did you try analyses like f_3 -statistics that explicitly test for admixture, rather than f_4 tests that assesses a more general "genetic affinity"? Or does qpAdm already include such f_3 -statistics when it concludes a group (e.g. AHG, AAF, ACF) is admixed? If so, can you include the f_3 tests that explicitly test for admixture? My understanding is that qpAdm is similar to TREEMIX, which does not use f_3 -statistics and often in practice infers admixture when it is not present and/or misses admixture that is present. In other words, how do we know the values in Tables S4-S9 are not reflecting a more general recent shared ancestry without involving admixture? It would be good to clarify these points -- i.e. what can be interpreted reliably from qpAdm inference.

We do not agree with the reviewer's perception of f_3 -statistics as an explicit test of admixture and f_4 -statistic as an assessment of general genetic affinity. Both statistics provide a formal statistical test of treeness. i.e. whether a relationship of given populations can be explained by a simple tree-like relationship or requires additional gene flows between branches (Patterson et al 2012 Genetics 192:1065). In many recent studies, a violation of treeness (i.e. gene flow) has been interpreted as admixture, largely equating admixture with gene flow. In this study, we also follow the same approach, but we have highlighted uncertainty in the demographic process that resulted in gene flow in our discussion (i.e. pulse-like vs. long-term continuous background gene flow).

Like most other archaeogenetic studies, we prefer D statistics to admixture f_3 to test treeness throughout this study because admixture f_3 statistics quickly lose their power when data is of insufficient quantity. That is because post-admixture drift in the target population, including artefactual drift due to limited sampling, adds a big positive number to the statistic where only significantly negative statistics are considered as a robust signal of gene flow. AHG is the most extreme case: with a single pseudo-diploid individual, all loci have allele frequency estimate either 0 or 1, which guarantees admixture f_3 statistic to be non-negative. In contrast, D statistics are insensitive to population-specific drift and thus provide a still robust but much more sensitive test of gene flow.

qpAdm is a generalization of f_4 statistics (which are essentially identical to D statistics), which intends to summarize multiple f_4 statistics into a single framework, and thus allows estimation of admixture coefficients and assessment of multi-way admixture models. In short, it tests if a linear combination of reference populations (proxies for the true ancestral sources) mimics the target population adequately to keep symmetry (i.e. treeness) against multiple outgroups: i.e. $f_4(\text{Outgroup}_1, \text{Outgroup}_2; \text{Model}, \text{Target})$ is around zero for all outgroup pairs provided. Therefore, it has no relationship with TreeMix, which is an automated search algorithm for the most likely population tree and gene flow edges on top of it.

For the specific case of AHG for which the reviewer raised questions, we started with the following D statistics: $D(\text{AHG}, \text{WHG}; \text{Natufian/Levant_N}, \text{Mbuti}) \geq 4.8 \text{ SE}$ and $D(\text{AHG}, \text{Natufian/Levant_N}; \text{WHG}, \text{Mbuti}) \geq 9.0 \text{ SE}$ (table S3). This is a robust proof that neither of the trees ((AHG, WHG), Natufian/Levant_N) or ((AHG, Natufian/Levant_N), WHG) fits the data. The third

possibility, that AHG forms a common outgroup to WHG and Natufian and thus is equally related to them, is a priori unlikely and also rejected by $D(\text{WHG}, \text{Natufian}; \text{AHG}, \text{Mbuti}) = 4.3 \text{ SE}$. These results show that it is necessary to include at least one gene flow event to explain the relationship between AHG, WHG and Natufian/Levant_N. We think it is most parsimonious to model an intermediate one (AHG) as a mixture of the two extremes (WHG and Natufian/Levant_N).

PCA and ADMIXTURE are descriptive methods that are prone to sample size, population-specific drift and data quality biases. Therefore, it is common to see that a naive and literal interpretation of PCA/ADMIXTURE results often leads to a biased estimate (e.g. please see Lawson et al (2018) "A tutorial on how not to over-interpret STRUCTURE/ADMIXTURE bar plots", Nature Communications 9, 3258). On the other hand, D statistics and qpAdm are insensitive to population-specific drift and to sample size, as we outlined above. We highlight that both methods still qualitatively support the same conclusion (i.e. AHG derives its ancestry from both WHG- and Natufian/Levant_N-related gene pools), while prefer to use admixture coefficient estimates from qpAdm for the above reasons.

As another example, one of the major points of the article is that there is genetic continuity between Anatolian Hunter Gatherers (AHG) and pre-Ceramic Anatolian Farmers (AAF). Your PCA (Fig 1) and ADMIXTURE (Fig S1) plots are consistent with this. However you also hint that there appears to be admixture (or "incoming ancestry") related to an Iranian Neolithic-like source in the AAF, which I think is less clear. While it is true that your plots (Fig 2A) and Fig S2 show a greater affinity between AAF and populations to the east relative to AHG, it is not clear this is due to admixture. First off, these admixture signals again seem inconsistent with your ADMIXTURE figure (Fig S1), which does not suggest that AHG and AAF are different (though maybe hinting that 1-2 AAF samples have very slightly increased Iran_N-like DNA). As a hypothetical example, if AAF were directly continuous from AHG without any outside ancestry contributions, could you get qpAdm values of inferred Iran_N ancestry similar to that you see in Table S6 if e.g. your AHG sample happened to come from a bottlenecked population that thus does not perfectly reflect the AHG population from which AAF derives? You also only have one AHG individual, which makes these analyses complicated and presumably prone to a lack of power, as suggested by your long confidence intervals when jack-knifing. A discussion and/or test of this uncertainty would be helpful.

We acknowledge that the signal related to additional Iranian/Caucasus ancestry is not prominent, probably due to its small contribution (only about 10%) as well as our limited sample size. Indeed, this might have resulted in similar ADMIXTURE profiles of AHG and AAF. However, we show that adding Neolithic Iranians as the second source (in addition to AHG) significantly increases the model fit to the data ($p = 0.014$; also please see our reply to reviewer #2). Admixture models replacing Neolithic Iranians with similarly old ancient individuals (e.g. Levant_N or WHG) significantly deviates from the AAF data (Table S6), suggesting that we have statistical power to distinguish between various sources. The hypothetical scenario described by the reviewer is unlikely to produce a false positive signal for admixture because D-statistics are insensitive to population-specific drift: in this scenario, our AHG and the source population of AAF still form a sister clade and thus no deviation of D-statistic from zero is expected when the true source is replaced by AHG.

For example, when testing for additional ancestry in AAF that is lacking in AHG, your null is that AHG and AAF genetically are a single population. So could you do a permutation test where you swap your single AHG with a single AAF sample and re-calculate $D(\text{AAF}^*, \text{AHG}^*; \text{test}, \text{Mbuti})$, where AHG* is this single AAF sample and AAF* now includes all other AAF plus the AHG (or perhaps simply excluding the AHG)? Try replacing AHG* with each of the AAF samples and note how often you get patterns similar to that in Fig 2A/Fig S2 (e.g. how often your maximum D value is greater than the max in Fig 2A, and/or how often this D score is positive when "test" comprises one of the eastern populations with positive D in Fig 2A). This would assess whether there really is a tendency for AHG to be less related to the eastern test groups, or whether by chance a single AAF individual might also have less affinity to these eastern groups relative to the other AAFs. Indeed your Supp data table S10 suggests the latter will be the case? In general it would help to run such tests, in addition to f-3 statistics to test explicitly for admixture, to get an idea of

certainty about these claims. It could be that only some AAF carry this N.Iran-like ancestry (i.e. as indicated in Fig S1) and/or that these signals can be explained by chance ancestry differences in a homogeneous population rather than genuine admixture. As mentioned above, could they also reflect a lack of perfect surrogates?

We performed the permutation test suggested by the reviewer (figure S3; data table S11; methods section lines 370-383) in which we alternate AHG with a different AAF individual in $D(\text{AAF}, \text{AHG}; \text{test}, \text{Mbuti})$ when in turn grouping the AHG individual with the other four AAF ones. We plot the results of these 5 tests and those of the original setting for the populations that had the 10 highest D-scores in the original test setting. As expected, all permuted settings are around zero within ± 1 standard errors while the original D-scores for these populations were more positive than all five permutations and resulted in a positive D-score ≥ 1 standard error.

You can perhaps do a similar permutation procedure to assess the reliability of claims about ACF having more ancient Levant-like ancestry relative to AAF -- e.g. what happens to Fig 2B and Fig S3 if you randomly permute AAF/ACF labels; do the affinities in Fig 2B and Fig S3 converge to 0 as expected?

We performed this suggested analysis as well. Leveraging over large enough sample sizes (5 AAF and 25 ACF), we produced 1,000 permutations of AAF/ACF population labels by randomly choosing five individuals for AAF and placing the rest into ACF. We performed this permutation for the four "test" populations which showed the most positive D statistic of the form $D(\text{ACF}, \text{AAF}; \text{test}, \text{Mbuti})$ with the original AAF/ACF population label (Levant_N, Natufian, Greece_EN and Balkans_Neolithic). We report the results in the form of histogram (figure S5; methods section lines 370-383). In all cases, the observed D statistic is within the top 0.5% tail, supporting the robustness of the observed signal.

Figure 3 is difficult to follow. First off, it would be helpful to re-acquaint the reader with the idea of Basal Eurasian ancestry, including tree plots possibly, and the intuition (again perhaps using figures in the SOM) behind the test you are doing here to infer the Basal Eurasian component (i.e. alpha). Again uncertainty is difficult to assess here. In particular it seems hard to believe that estimating ~0% ancestry in Iron Gates HS is inconsistent with the truth being ~6% -- are estimates really that precise, even though you're using only a single AHG sample? Some discussion on this, or some testing using simulations, would be helpful, because it is difficult to understand how strong a statement can be made here.

We acknowledge that the concept of "Basal Eurasian ancestry" might not be familiar to many readers and accordingly added a section introducing this concept and the logic of our use of it as a tracer for a gene flow from ancient Near Eastern gene pool (please see our response to reviewer #1). In the revised manuscript, we clarify our logic: while no Basal Eurasian ancestry was required to model Iron Gate HGs, an alternative model using AHG as a reference implies $6.4 \pm 1.9\%$ Basal Eurasian ancestry in Iron Gate HGs (multiplying 25.8% AHG ancestry proportion to Basal Eurasian ancestry proportion in AHG, $24.8 \pm 5.5\%$). Although 6.4% is a small quantity, it is more than three standard errors above zero. Based on this result, we suggest that one-way gene flow from AHG into Iron Gate HGs is not enough to explain a strong affinity between them. Acknowledging limited resolution of our data, we describe several additional scenarios that cannot be distinguished using currently available data (such as a bidirectional gene flow or one directional gene flow from the ancestors of Iron Gates to those of AHG). In the discussion section we note that the spatiotemporal extent of the genetic interactions is not yet clear and requires further sampling (p. 12, line 267). We modified the text to highlight these arguments and to clarify our statements (p. 12, lines 256-267).

minor comments

-- throughout you report ancestry percentages to the 10th decimal place -- can it really be this precise?
For the sake of consistency and reproducibility we provided the values as they were rounded when outputted by our analyses tools.

Fig S1 -- there appears to be little difference between K=6-15 here, presumably in part because you are running ADMIXTURE on a world-wide collection of samples (Fig S5), most of which are not relevant for what you are studying here. Can you run ADMIXTURE on only samples relevant to this analysis, e.g. excluding Africa and other regions? This might increase power to assess whether there are genetic differences between AAF, ACF, AHG, etc, and how these groups relate to ancient Levantines, Iranian Neolithics, etc.

Following the reviewer's suggestion, we attempted an ADMIXTURE run including western Eurasians only. However, the results seem to be noisier rather than more informative. We believe that our world-wide ADMIXTURE results with high enough K values capture population structure within ancient and modern western Eurasians to the level sufficient for generating hypotheses we formally tested in our analysis and thereby decided not to present western Eurasian ADMIXTURE results.

SOM references -- The numbers are higher (e.g. 56, 57) than what's listed in these sections, so need to be re-set presumably?

We appreciate the reviewer for pointing out typos. We fixed the issue in the revised manuscript.

Reviewers' comments:

Reviewer #1 (Remarks to the Author):

My concerns have been satisfactorily dealt with.

Note one reference typo in the Methods section - Fu et al is not reference 30.

Reviewer #2 (Remarks to the Author):

The authors have appropriately addressed my comments. I have no further issues with the manuscript.

Reviewer #3 (Remarks to the Author):

Regarding your comment about f3 versus f4 tests, the former "definitely shows that a history of mixture occurred" in the target population (p.1068 of Patterson et al 2012, Genetics). In contrast, f4/D-stats are general tests of treeness, and which groups give the most positive/negative scores depends on levels of drift in those groups. However, as you point out, cases where you only have a single call per SNP to define a "population" render the f3 admixture test unusable. So it does make sense to rely on f4-stats and (in particular) qpAdm in such scenarios. (I have looked into the latter, and now understand it is not like TREEMIX and seems a more robust test of admixture; thank you for clarifying.) However, it would be helpful for readers if you state this reasoning for not using f3 in the text (e.g. S13), and to provide the brief description of the more complicated qpAdm model that you provided to me. Also it would be helpful to reiterate the assumptions of the methods you are using for unfamiliar readers, such as how qpAdm assumes the outgroups have not experienced recent gene flow with the reference or test populations. One source of confusion is that the S13 text suggests you could have used Natufian as a reference even though you have defined it as an outgroup; presumably it cannot be both?

The permutation tests for Fig S3 are helpful. As is, it seems to suggest that caution is warranted here. The Luk9 test suggests that a single AAF can indeed show greater affinity to eastern groups relative to the other AAF samples that are meant to be entirely homogeneous, in an analogous manner to how AHG shows less affinity to eastern groups. However, a caveat is that you included AHG among the AAF*, which may be exacerbating the difference. So it is worth redoing this test of Luk9 without AHG. After removing AHG, if the line is still similarly shifted down (for e.g. the Mala comparison), this could reflect Luk9 carrying more such Iran_N-like admixture (i.e. if such gene flow is recent), though it could just as well reflect your single AHG sample carrying less such admixture (or possibly no admixture in any samples). This challenge should be pointed out somewhere if it persists when excluding AHG (e.g. S13 and likely a sentence of caution in the main text, p.6), as this test does not rely on the modeling assumptions that qpAdm does. It would also be helpful to explain briefly in the Methods section (or S13) why you are doing these permutations, in addition to how you did them.

The Fig S5 permutation is also helpful, providing further evidence that AAF and ACF clearly are not exchangeable in regards to their amounts of West-Eurasian ancestry.

I think your justification for not including the ADMIXTURE plots on only your samples of interest is okay, but you should state in the text (or SI) that you tried it and the results were noisier, i.e. still without any clear delineations between AAF/ACF/AHG.

Point by point response to reviewer's comments

Reviewer #1 (Remarks to the Author):

My concerns have been satisfactorily dealt with.

Note one reference typo in the Methods section - Fu et al is not reference 30.

*We thank the reviewer for their contribution to our paper.
The typo has been corrected in the revised manuscript (p13, line 32).*

Reviewer #2 (Remarks to the Author):

The authors have appropriately addressed my comments. I have no further issues with the manuscript.

We thank the reviewer for their contribution to our paper.

Reviewer #3 (Remarks to the Author):

Regarding your comment about f3 versus f4 tests, the former "definitely shows that a history of mixture occurred" in the target population (p.1068 of Patterson et al 2012, Genetics). In contrast, f4/D-stats are general tests of treeness, and which groups give the most positive/negative scores depends on levels of drift in those groups. However, as you point out, cases where you only have a single call per SNP to define a "population" render the f3 admixture test unusable. So it does make sense to rely on f4-stats and (in particular) qpAdm in such scenarios. (I have looked into the latter, and now understand it is not like TREEMIX and seems a more robust test of admixture; thank you for clarifying.) However, it would be helpful for readers if you state this reasoning for not using f3 in the text (e.g. SI3), and to provide the brief description of the more complicated qpAdm model that you provided to me.

We thank the reviewer for their additional comments.

As the reviewer suggested, we include in our revised manuscript our reasoning for using D-statistics: "D-statistics provide a robust and sensitive test of gene flow and are preferable for low quantity data analysis (typical of Archaeogenetic studies) as they are insensitive to post-admixture drift, including artefactual drift due to a limited sample size"⁴³ (p. 16 lines 370-372).

We would like to emphasize again that D-statistics are as a robust and sensitive test of admixture as f3-statistics but unlike f3-statistics have the advantage that they can be used for small sample sizes and are insensitive to population specific drift. To quote from the same paper referenced above: "the D-statistics show a substantial deviation from 0 only when an admixture event occurred in the history of the four populations contributing to the statistic" (p.1073 of Patterson et al 2012, Genetics).

Following the reviewer's request, we now provide a brief description of qpAdm in the revised main text (pp. 5-6, lines 124-129).

Also it would be helpful to reiterate the assumptions of the methods you are using for unfamiliar readers, such as how qpAdm assumes the outgroups have not experienced recent gene flow with the reference or test populations. One source of confusion is that the SI3 text suggests you could have used Natufian as a reference even though you have defined it as an outgroup; presumably it cannot be both?

Indeed, the qpAdm outgroups were used only if they are unlikely to harbor recent gene flow with the target or references. We now explain this prerequisite in the revised SI: "These outgroups were chosen to distinguish the ancestry of the reference populations since they broadly represent the known global genetic diversity and are unlikely to harbor recent gene flow with the target or reference populations either due to geographical/temporal distance or based on their genetic clustering in ADMIXTURE and PCA analysis" (SI3, p23 lines 14-16). To avoid confusion, we also removed the sentence regarding potentially using "Natufian" as a reference in the revised SI3.

The permutation tests for Fig S3 are helpful. As is, it seems to suggest that caution is warranted here. The Luk9 test suggests that a single AAF can indeed show greater affinity to eastern groups relative to the other AAF samples that are meant to be entirely homogeneous, in an analogous manner to how AHG shows less affinity to eastern groups. However, a caveat is that you included AHG among the AAF*, which may be exacerbating the difference. So it is worth redoing this test of Luk9 without AHG. After removing AHG, if the line is still similarly shifted down (for e.g. the Mala comparison), this could reflect Luk9 carrying more such Iran_N-like admixture (i.e. if such gene flow is recent), though it could just as well reflect your single AHG sample carrying less such admixture (or possibly no admixture in any samples). This challenge should be pointed out somewhere if it persists when excluding AHG (e.g. SI3 and likely a sentence of caution in the main text, p.6), as this test does not rely on the modeling assumptions that qpAdm does. It would also be helpful to explain briefly in the Methods section (or SI3) why you are doing these permutations, in addition to how you did them.

We preformed the requested analysis excluding AHG and added the results to Fig. S3. As expected the D-statistics become less negative when Luk9 is compared only with the remaining AAF individuals (-2.6 to -0.8 SE with AHG vs. -1.9 to -0.3 SE without AHG). In fact, all possible within-AAF comparisons (i.e. one AAF vs the remaining four AAFs) show $|D| < 1.9$ SE, while the original comparisons between AHG and AAF show stronger signals of differentiation ($D = +1.4$ to $+2.7$ SE; 9 out of 10 D statistics > 2 SE). We believe that these permutation results support the followings: i) all five AAF individuals have more genetic affinity with CHG/Iran_N related ancestry than AHG does, and ii) such affinity is less variable within AAF individuals than it is between AAF and AHG. Indeed, signals less than 2 SE are typically not considered even as suggestive evidence of difference.

We acknowledge that our limited sample size does not allow us to robustly test whether the AHG and AAF individuals form largely disjoint distributions. However, we would like to point out that it is an extreme condition of population differentiation rather than the minimal condition. D- statistic tests mean difference between two groups, which can be significantly different even if two distributions partially overlap, while a permutation test is highly significant only when two groups have non-overlapping distributions. We look forward to conducting future studies to increase sample size in ancient Anatolia, and accordingly highlighted the importance of further sampling in the previous version of the manuscript (p. 12

lines 272-273). Following the reviewer's suggestion, we also added in the revised manuscript a sentence highlighting this issue with regards to this specific analysis (SI3, p25 lines 5-8). Also, we added a sentence explaining our motivation to perform the permutation tests to the revised methods section (p16 lines 380-388).

The Fig S5 permutation is also helpful, providing further evidence that AAF and ACF clearly are not exchangeable in regards to their amounts of West-Eurasian ancestry.

Thank you for your positive reception of the permutation results.

I think your justification for not including the ADMIXTURE plots on only your samples of interest is okay, but you should state in the text (or SI) that you tried it and the results were noisier, i.e. still without any clear delineations between AAF/ACF/AHG.

We added the requested statement to the revised manuscript (p.16 lines 363-365).

REVIEWERS' COMMENTS:

Reviewer #3 (Remarks to the Author):

My reviews have been addressed satisfactorily -- no further comments.